# Co-ultraPEALut in Subjective Cognitive Impairment Following SARS-CoV-2 Infection: An Exploratory Retrospective Study

**DOI:** 10.3390/brainsci14030293

**Published:** 2024-03-20

**Authors:** Valentina Cenacchi, Giovanni Furlanis, Alina Menichelli, Alberta Lunardelli, Valentina Pesavento, Paolo Manganotti

**Affiliations:** 1Clinical Unit of Neurology, Department of Medicine, Surgery and Health Sciences, University Hospital and Health Services of Trieste—ASUGI, University of Trieste, Strada di Fiume, 447, 34149 Trieste, Italy; giovanni.furlanis@asugi.sanita.fvg.it (G.F.); paolo.manganotti@asugi.sanita.fvg.it (P.M.); 2Neuropsychological Service, Clinical Unit of Rehabilitation, University Hospital and Health Services of Trieste, ASUGI, 34125 Trieste, Italy; alina.menichelli@asugi.sanita.fvg.it (A.M.); alberta.lunardelli@asugi.sanita.fvg.it (A.L.); valentina.pesavento@asugi.sanita.fvg.it (V.P.)

**Keywords:** long-COVID-19, neuro-long-COVID-19, SARS-CoV-2, cognitive impairment, co-ultraPEALut

## Abstract

Neurological involvement following coronavirus disease 19 (COVID-19) is thought to have a neuroinflammatory etiology. Co-ultraPEALut (an anti-inflammatory molecule) and luteolin (an anti-oxidant) have shown promising results as neuroinflammation antagonists. The aim of this study was to describe cognitive impairment in patients with post-COVID-19 treated with co-ultraPEALut. The Montreal Cognitive Assessment (MoCA), the Prospective–Retrospective Memory Questionnaire (PRMQ), the Fatigue Severity Scale (FSS), and a subjective assessment were administered at baseline and after 10 months. Patients treated with co-ultraPEALut were retrospectively compared with controls. Twenty-six patients treated with co-ultraPEALut showed a significant improvement in PRMQ (T0: 51.94 ± 10.55, T1: 39.67 ± 13.02, *p* < 0.00001) and MoCA raw score (T0: 25.76 ± 2.3, T1: 27.2 ± 2, *p* 0.0260); the MoCA-adjusted score and the FSS questionnaires also showed an improvement, even though it was not statistically significant; and 80.77% of patients reported a subjective improvement. In the control subjects (*n* = 15), the improvement was not as pronounced (PRMQ T0: 45.77 ± 13.47, T1: 42.33 ± 16.86, *p* 0.2051; FSS T0: 4.95 ± 1.57, T1: 4.06 ± 1.47, *p* 0.1352). Patients treated with co-ultraPEALut and corticosteroids were not statistically different from those treated with co-ultraPEALut alone. Neuro-post-COVID-19 patients treated with co-ultraPEALut scored better than controls in MoCA and PRMQ questionnaires after 10 months: this may support the importance of neuroinflammation modulation for neuro-long-COVID-19.

## 1. Introduction

Severe acute respiratory syndrome coronavirus 2 (SARS-CoV-2) first reported in Wuhan, China in December 2019 and subsequently spread worldwide. As of March 2024, the number of confirmed SARS-CoV-2 cases is over 774 million, according to the World Health Organization’s COronaVIrus Disease 19 (COVID-19) Dashboard. COVID-19 does not resolve its clinical relevance in the acute phase of the disease but can trigger a nosological entity known as post-COVID-19 syndrome or also “long COVID”, “persistent COVID”, “long-tail COVID” and “post-acute sequelae of SARS-CoV-2” (PASC) [1]. The National Institute for Health and Care Excellence (NICE) defined post-COVID-19 syndrome as “signs and symptoms that develop during or after an infection consistent with COVID-19, last longer than 12 weeks and cannot be explained by an alternative diagnosis” [2]. The US Centers for Disease Control and Prevention (CDC) and the National Institutes of Health (NIH) have adopted a slightly narrower definition that only considers symptoms that persist for more than 4 weeks after the initial infection to be the long COVID-19 syndrome [3]. Either way, as post-COVID-19 is characterized by relevant neurological involvement together with frequent respiratory and cardiovascular symptoms, the umbrella term “neuro-long-COVID” has been proposed [4]. 

Whether neuro-long-COVID-19 should be considered a distinct pathological entity or not is controversial: some authors claim that its neurological signs and symptoms are poorly characterized and lack consistency, thus resembling functional disorders [5], while other authors openly oppose calling the neurological involvement in post-COVID-19 functional [6]. Be that as it may, direct SARS-CoV-2 invasion into the central nervous system, abnormal systemic and neurological immunological responses, cytokine storm, glial dysfunction, virosis-induced endothelial changes and other mechanisms have been proposed as pathogenetic mechanisms underlying post-COVID-19 neurological symptoms [7]. Of these mechanisms, neuroinflammation emerges as a likely pathogenetic element: even before the spread of SARS-CoV-2, neuroinflammation was studied as a physiopatogenetic factor in some neuropsychiatric disorders such as major depression, in which the level of immune activation is directly related to cognitive–behavioral changes [8]. In this context, serum biomarkers of inflammation such as C-reactive peptide, D-dimer, proinflammatory cytokines, procalcitonin, Vascular Cell Adhesion Molecule-1 (sVCAM-1) and components of the complement pathway have been demonstrated to be significantly elevated in patients with neurological involvement after COVID-19 [1,9].

Therefore, it is important to find new approaches to control the neuroinflammatory process at multiple levels. In this regard, the co-ultramicronization of palmitoylethanolamide (PEA) and luteolin, known as co-ultraPEALut, could represent a possible intervention: it has shown promising results in preclinical studies as a neuroinflammation antagonist in diseases such as traumatic brain injury, vascular dementia and ischemic stroke [10,11,12], as well as in clinical studies on post-COVID-19-persistent olfactory impairment [13,14] and post-COVID-19 memory impairment [15]. PEA is an endogenous molecule belonging to the N-acylethanolamine family, found in several human tissues, including the brain, and is synthesized “on demand” in response to stressors to restore tissue homeostasis. It exerts its protective effect thanks to its ability to modulate the hyperactivation of mast cells through the so-called autacoid local injury antagonism (ALIA) mechanism [16]. Numerous publications have documented the efficacy of PEA in various diseases that have different etiologies but share the pathogenetic mechanism of non-resolving neuroinflammation. Moreover, it has been demonstrated that co-ultramicronization between umPEA (the ultramicronized form of PEA, which increases its bioavailability and thus its biological efficacy) [17,18] and specific polyphenols such as polydatin or luteolin produces microcomposites with higher neuroprotective, anti-inflammatory and anti-oxidant properties [19,20,21]. Luteolin is a plant polyphenol flavonoid that has shown the ability to downregulate the production of tumor necrosis factor alpha, interleukins and free radicals and has been proven to decrease vascular permeability when increased by inflammatory processes. Overall, it has anti-inflammatory, anti-oxidant and cytoprotective properties [22,23,24,25].

Other molecules have been proposed for the treatment of post-COVID-19 memory impairment through the putative anti-neuroinflammatory mechanisms, such as corticosteroids, as they stabilize mast cells and activate brain microglia in addition to suppressing the production of pro-inflammatory cytokines [26].

The aim of this study was (i) to describe the course of cognitive impairment in post-COVID-19 patients treated with co-ultraPEALut using neuropsychological tests to evaluate overall cognitive status, memory and fatigue (using the Montreal Cognitive Assessment [MoCA] test, the Prospective and Retrospective Memory Questionnaire [PRMQ] and the Fatigue Severity Scale [FSS], respectively) and (ii) to compare the outcomes of patients treated with and without co-ultraPEALut.

## 2. Materials and Methods

The present retrospective study involved 26 outpatients suffering from subjective cognitive impairment occurring after SARS-CoV-2 infection and treated at the Neurology Department of the University Hospital and Health Service of Trieste (Italy) between July 2021 and December 2022. Patients of both genders with a previous SARS-CoV-2 infection documented by a positive nasopharyngeal swab and who reported subjective cognitive decline (SCI) during and/or after the infection for a period of at least 12 weeks were considered for this study. SCI, also known as “brain fog”, is primarily a clinical diagnosis that refers to the self-reported deterioration of abilities in one or more cognitive domains, such as concentration, memory, reasoning or problem solving. There are no detailed cutoffs to meet; in order to diagnose SCI, it is sufficient that the patient states an impairment of their cognitive abilities when compared to previous habitual functioning. Patients with a previous diagnosis of major cognitive or psychiatric disorders (e.g., dementia, schizophrenia, bipolar disorder), ongoing corticosteroid therapy at the time of the initial neurological evaluation or with a more likely explanation for the neurological deficits after the diagnostic work-up were excluded from the study. As, on the other hand, depressive and/or anxious traits are common among patients with post-COVID-19 neurological involvement, the participants in our study were screened with the Symptom Checklist-90-Revised (SCL-90-R). The SCL-90-R is a 90-item self-report symptom inventory that measures psychological symptoms and distress; each element provides a score on a Likert scale ranging from 0 (no perceived distress) to 4 (maximum subjective distress experience). In particular, we chose to report depression (DEP) and anxiety (ANX) dimensions.

The diagnostic work-up included a clinical assessment by a neurologist dedicated to post-COVID-19 neurological involvement, blood tests, cerebral imaging (preferably magnetic resonance imaging, MRI) and electroencephalography. The clinical assessment combined history taking with neuropsychological assessment, which included the following:i.Montreal Cognitive Assessment test (MoCA), adjusted for age and educational level according to Aiello [27,28]: one of the most widespread and robust screening tests to evaluate both non-instrumental (executive functions, attention) and instrumental (language, memory, visuospatial abilities, orientation) cognitive domains. It is scored out of 30 points, with higher scores indicating better performance. A cutoff of 24 points is generally used for normality. To avoid a learning effect, we used alternate forms of the Italian version of MoCA for the two serial evaluations at the beginning and end of the study. Remarkably, MoCA proved to be superior to the Mini-Mental State Examination (MMSE) when it came to detecting subclinical defects in post-COVID-19 patients [29].ii.Fatigue Severity Scale (FSS) [30]: a questionnaire that scores the average of 9 items, each ranging from 1 (strongly disagree) to 7 (strongly agree), with a cutoff score >4 indicating significant fatigue. It is a valid and reliable self-report instrument to measure fatigue, defined as a significantly diminished energy level and/or an increased perception of effort [31].iii.Prospective and Retrospective Memory Questionnaire (PRMQ) [32]: a self-report rating scale designed to determine the frequency of prospective and retrospective memory difficulties in everyday life consisting of sixteen items scored on a 5-point scale (5 = very often; 1 = never), eight asking about prospective memory failures and eight concerning retrospective failures, with higher scores indicating a more frequent occurrence of memory errors or memory loss.iv.Subjective assessment of health changes, which allowed each patient to be categorized as worsened, stable, improved or remitted compared to the previous visit.

Patients were treated with a commercial oral formulation consisting of a co-ultramicronized association of 700 mg PEA and 70 mg luteolin (Food for Special Medical Purposes, Glialia^®^, Epitech Group SpA, Saccolongo, Italy), for a period of at least two months (twice daily for 30 days and once daily for the following 30 days or more), as an add-on to their concomitant therapy for comorbidities.

All the above-mentioned examinations, except for the subjective assessment proposed at the end of the study, were performed before the start of treatment with co-ultraPEALut (T0) and at the time of re-evaluation (T1) after an average of almost 10 months, according to clinical practice in the participating outpatient unit.

Previously collected data from neuro-long-COVID-19 patients who were not treated with co-ultraPEALut (15 control patients) were used to compare MoCA, FSS and PRMQ scores at baseline and mean FSS and PRMQ scores between the two groups before and after treatment. A secondary exploratory analysis was performed on a small subgroup of patients treated with co-ultraPEALut who were given corticosteroids (prednisone) orally at a dose of 1 mg/kg/die for 3 days followed by an 8-day décalage, in addition to the aforementioned therapies. In these patients, the administration of co-ultraPEALut and corticosteroids was started simultaneously.

Patients receiving corticosteroids were warned about possible adverse or side effects of this type of medication, as in routine clinical practice. At the time of re-evaluation, all patients were asked to report any adverse effects of therapy (both co-ultraPEALut and corticosteroids).

We performed all statistical analyses using SPSS Statistics 22 (IBM, Armonk, NY, USA). The data obtained were tested for normality using the Kolmogorov–Smirnov test and then compared using Student’s *t*-test. Values are expressed as mean ± standard deviation (SD). A *p*-value of less than 0.05 was considered statistically significant.

The study was conducted in accordance with the principles of the Declaration of Helsinki and good clinical practice (GCP).

## 3. Results

### 3.1. Patients’ Characteristics

Twenty-six outpatients (73.08% females) treated with co-ultraPEALut were evaluated (Table 1, Column 1): their mean age ± S.D. was 54.69 ± 11.26 years; their mean level of education was 14.10 ± 3.05 years; 21 of them (80.77%) were employed; 1 (3.85%) was unemployed; and 3 (15.38%) were retired. In 22 of these patients (84.62%) a paucisymptomatic infection occurred; 4 (15.38%) required hospitalization, and none had to be admitted to the intensive care unit. The time between the first nasal swab that tested positive for SARS-CoV-2 (NS) and the first neurological assessment (T0) was 7.33 ± 4.75 months, while the time between T0 and reassessment (T1) was 9.92 ± 5.37 months.

Data from 15 patients not treated with co-ultraPEALut (Table 1, Column 2) were collected and used as controls: they yielded similar demographic characteristics, except for the ratio of female to male patients, which was lower in the control group. Sex comparisons were not specifically sought in this study, as previous research on the neuro-long-COVID-19 outpatients at our clinic showed no significant differences between female and male symptoms [4].

### 3.2. Neuropsychological Outcomes

At the time of the first assessment, raw and adjusted MoCA, PRMQ and FSS scores were comparable between co-ultraPEALut and control patients, with PRMQ scores being only slightly lower in the control group. Mean scores on the depression (DEP) and anxiety (ANX) dimensions of the Symptom Checklist-90-Revised (SCL-90-R) questionnaire were also similar at T0 between patients treated with co-ultraPEALut and controls: the *t*-test for independent means on the comparison between groups yielded a *p* 0.7948 for DEP scores and a *p* 0.9849 for ANX scores, meaning that depression and anxiety scores at baseline were comparable between the two groups (Table 2, Figure 1).

#### 3.2.1. MoCA

In the group treated with co-ultraPEALut, MoCa raw scores statistically significantly increased (i.e., improved) between the first assessment and the reassessment, with *p* 0.0260 (T0: 25.76 ± 2.3, T1: 27.2 ± 2); a trend toward improvement was noted for MoCA-adjusted scores with *p* 0.0958 (T0: 24.34 ± 2.58, T1: 25.59 ± 2.39) (Table 2, Figure 1). Unfortunately, no comparative analysis could be performed for MoCA scores in the control group, as this test was only administered at T0 to the patients not treated with co-ultraPEALut.

As a secondary analysis, a comparison was made between the subgroup of patients treated with co-ultraPEALut only (19 patients) and the group treated with both co-ultraPEALut and corticosteroids (7 patients). As the numerosity of these two subgroups is uneven and small, results are to be taken cautiously. MoCA raw scores increased in the subsample receiving co-ultraPEALut only (T0: 25.93 ± 2.52, T1: 27.11 ± 1.82, *p* 0.1417) and in that receiving co-ultraPEALut and corticosteroids in combination (T0: 25.33 ± 1.21, T1: 27.43 ± 2.3, *p* 0.0709); similarly, the MoCA-adjusted score increased in the former (T0: 24.54 ± 2.82, T1: 25.33 ± 2.03, *p* 0.8488) and the latter subsample (T0: 24.47 ± 2.2, T1: 26.27 ± 3.11, *p* 0.2609), respectively (Table 3). On average, raw and adjusted MoCA scores increased by 1.18 and 0.79 points in the group treated with co-ultraPEALut and by 2.1 and 1.8 points in the group treated with co-ultraPEALut plus corticosteroids; a *t*-test performed on the mean of the differences from first and second assessment between the two subgroups yielded a *p* of 0.3081 and 0.4875, respectively.

#### 3.2.2. FSS

In the co-ultraPEALut-treated group, FSS scores showed a reduction (i.e., improvement) between the first and the second neurological assessment, although it was not statistically significant (T0: 5.03 ± 1.69, T1 4.52 ± 1.63, *p* 0.3188). In the control group, FSS scores underwent a similar reduction (T0: 4.95 ± 1.57, T1: 4.06 ± 1.47, *p* 0.1352) (Table 2, Figure 1). On average, FSS scores decreased from T0 to T1 by 0.51 points in the co-ultraPEALut-treated group and by 0.89 points in the control group.

The mean FSS score decreased in the subgroup treated with co-ultraPEALut alone (T0: 4.54 ± 1.54, T1: 4.29 ± 1.68, *p* 0.6803) and in the subgroup treated with both co-ultraPEALut and corticosteroids (T0: 6.8 ± 0.23, T1: 5.24 ± 1.32, *p* 0.6251) (Table 3); a *t*-test performed on the mean of the differences from first and second assessment between the two subgroups yielded a significant *p* of 0.01754 (co-ultraPEALut alone: 0.25 points; co-ultraPEALut and corticosteroids: 1.56 points).

#### 3.2.3. PRMQ

In the co-ultraPEALut-treated group, PRMQ scores reached a statistically significant reduction (i.e., improvement) with *p* < 0.00001 (T0: 51.94 ± 10.55, T1: 39.67 ± 13.02). Control patients, on the other hand, showed a milder, not statistically significant, reduction in PRMQ scores (T0: 45.77 ± 13.47, T1: 42.33 ± 16.86, *p* 0.2051) (Table 2, Figure 1). On average, PRMQ scores decreased by 12.27 points in the co-ultraPEALut-treated group and by 7.97 points in the control group.

PRMQ score reduction was statistically significant, with a *p* 0.0067, in patients treated with co-ultraPEALut alone (T0: 53.38 ± 9.92, T1: 40.24 ± 13.3), while in patients treated with co-ultraPEALut and corticosteroids it was not (T0: 47.25 ± 11.35, T1: 38.29 ± 12.88, *p* 0.2782) (Table 3); a *t*-test performed on the mean of the differences from first and second assessment between the two subgroups yielded a *p* of 0.3662 (co-ultraPEALut alone: 13.14 points; co-ultraPEALut and corticosteroids: 8.96 points).

#### 3.2.4. Subjective Assessment of Change in Health Status

Information about the subjective assessment of change in health status over time was collected at the time of the second neurological assessment only, due to the nature of these data. Of the patients treated with co-ultraPEALut, four (15.38%) reported stability in their health status, sixteen (61.54%) perceived improvement and five (19.23%) reported complete remission of symptoms with a return to baseline health status, while only one patient reported an overall worsening of their symptoms (Table 2, Figure 2).

With regard to our secondary analysis on the concomitant administration of corticosteroids, the subjective assessment of change in health status over time did not differ significantly in the two subgroups of patients: among subjects receiving only co-ultraPEALut, three (15.79%) reported stability in their health status, twelve (63.16%) reported improvement, three (15.79%) reported complete remission of symptoms, and only one (5.26%) reported worsening of their condition; among subjects receiving corticosteroids and co-ultraPEALut, the outcomes of stability, improvement, remission and worsening were reported by one (14.29%), four (57.14%), two (28.57%) and zero patients, respectively (Table 3, Figure 3).

### 3.3. Safety Outcomes

We did not expect any adverse effects from treatment with co-ultraPEALut as previous data show excellent safety results [10]. At the time of re-evaluation, none of the twenty-six patients receiving co-ultraPEALut reported any adverse safety findings related to treatment; of the seven patients receiving corticosteroids in addition to co-ultraPEALut, two reported mild side effects (mild sleep disturbance in one case, mild facial redness in the other), both of which resolved after the end of treatment.

## 4. Discussion

This exploratory retrospective study examined outpatients who were referred to our neurology department due to subjective cognitive impairment after a documented SARS-CoV-2 infection. Neuropsychological tests were performed before and after treatment with co-ultraPEALut, a substance whose anti-inflammatory properties had been demonstrated in both preclinical and clinical studies. Post-COVID-19 neurologically affected patients treated with co-ultraPEALut and observed over an average course of almost 10 months showed both objective and subjective cognitive improvement at the end of the observation period, with statistically significant results in the MoCA (raw score) and PRMQ questionnaires. Control patients who did not receive co-ultraPEALut also showed an improvement in cognitive function, albeit not as marked (e.g., in PRMQ scores). Furthermore, the exploratory comparison between patients treated with co-ultraPEALut alone and those treated with co-ultraPEALut and corticosteroids showed similar results, except for the severity of fatigue, as FSS seemed to decrease more in the latter group compared to the former; however, the interpretations of this result must be made with caution, as the numbers in the subgroups are small and the analysis is secondary. This may favor the pro-inflammatory pathophysiological hypothesis of neuro-long-COVID-19 and may highlight the importance of neuro-inflammation modulation in this type of neurological issue.

Indeed, COVID-19 patients show elevated pro-inflammatory serum and cerebro-spinal fluid (CSF) biomarkers, neuronal damage and glial activation that quantitatively correlate with neurological involvement and disease severity [33,34]. Among the most studied pro-inflammatory mediators in post-COVID-19 patients, both TNF-alpha, which is associated with impaired synaptic plasticity, microglial activation and subsequent detrimental effects on memory [35], and IL-6, which is associated with cognitive dysfunction in mouse models [36], are associated with mood changes and cognitive deficits [33].

Clinically, the most commonly reported neuro-long-COVID-19 symptoms are cognitive deficits (including brain fog), fatigue and persistent hypo-/anosmia; together with headaches, sleep disturbance, depression and anxiety as well as autonomic nervous system dysfunctions such as orthostatic intolerance and sudomotor changes [1,3,4,37]. A prospective study carried out in Milan (Italy) on 226 post-COVID-19 patients reported that 78% of them had difficulties in at least one cognitive domain, especially executive functions and motor coordination, and that 36% of them experienced psychological-psychiatric symptoms, e.g., from depression, 3 months after hospital discharge; moreover, a positive correlation between depressive symptoms and systemic inflammatory biomarkers during and after COVID-19 was demonstrated [38]. Furthermore, a study conducted in our clinic showed that 61% of patients reported autonomic nervous system dysregulation after COVID, which appears to be associated with pro-inflammatory immune dysregulation, as sympathetic activation induces cytokine production [37,39].

Patients with neuro-long-COVID-19 have also been studied using magnetic resonance imaging (MRI), and significant changes in cerebral perfusion have been found in arterial spin labeling sequences (ASL-MRI): the connectivity of the large network appears to be disrupted, as the metabolism of frontal, temporal and parietal areas appears to be altered [40]. These observation align with the metabolic abnormalities seen in several brain regions in 18FDG-PET studies [41]. In addition, electroencephalographic (EEG) features were abnormal with slowing or epileptiform discharges in two thirds of patients with neuro-long-COVID-19 [42]. Also, inhibitory GABAergic and excitatory glutamatergic regulatory circuits appear to be impaired in paired-pulse transcranial magnetic stimulation (ppTMS) with long-interval intracortical inhibition (LICI) and intracortical facilitation (ICF), which may be lower in people with neuro-long COVID, possibly reflecting a mild and often transient encephalopathy caused by the direct or indirect effects of SARS-CoV-2 [43].

Limitations of this study include the monocentric nature, lack of case–control matching, incomplete data and different examiners involved in the testing of co-ultraPEALut-treated and control patients. Moreover, because of lack of randomization and blinding, the results may reflect placebo effects rather than (or, at least, in combination with) true treatment effects. In the end, because our study did not achieve adequate statistical power, we cannot draw definite conclusions about the efficacy of co-ultraPEALut on subjective cognitive impairment following SARS-CoV-2 infection; however, what we found may have exploratory value for future research, possibly uncovering neural mechanisms associated with this treatment. This paper also opens the way to some interesting future perspectives, including the analysis of the cerebral MRI dataset and its correlation with clinical and neuropsychological data, possibly with follow-up MRI imaging.

## 5. Conclusions

Our main finding is that outpatients with subjective cognitive impairment after SARS-CoV-2 infection improved overall over time, both in neuropsychological assessment and, subjectively, after treatment with co-ultraPEALut. These results may provide preliminary evidence of the potential beneficial effect of this combination of anti-inflammatory and anti-oxidant molecules on neuro-post-COVID-19. Further research is warranted to elucidate the possible effect of co-ultraPEALut, alone or in combination with anti-inflammatory drugs, on neurological involvement in post-COVID-19.

## Figures and Tables

**Figure 1 brainsci-14-00293-f001:**
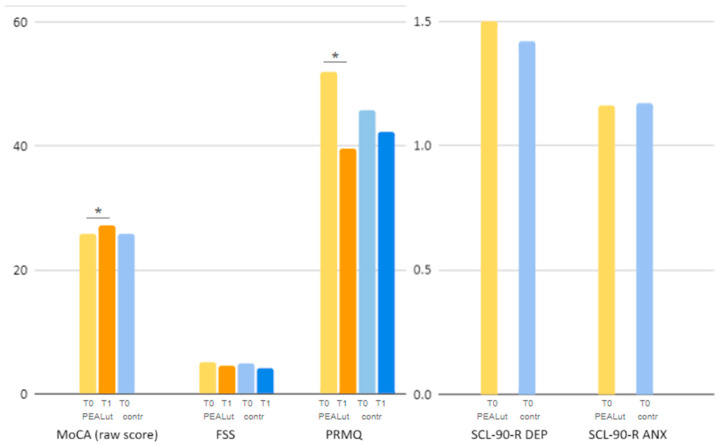
Visual representation of outcome measures comparison between patients receiving co-ultraPEALut and the control sample. Mean data are shown. FSS: Fatigue Severity Scale. MoCA: Montreal Cognitive Assessment test. PRMQ: Prospective and Retrospective Memory Questionnaire. T0: first neurological evaluation time. T1: re-evaluation time. SCL-90-R: Symptom Checklist-90-Revised questionnaire. DEP: depression dimension of the SCL-90-R questionnaire. ANX: anxiety dimensions of the SCL-90-R questionnaire. *: statistical significance for *p* < 0.05.

**Figure 2 brainsci-14-00293-f002:**
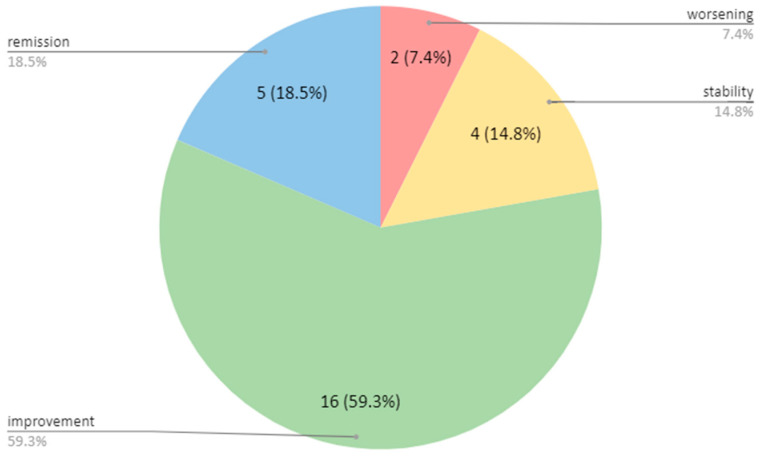
Subjective assessment of health change over time in the co-ultraPEALut group. Data displayed as %.

**Figure 3 brainsci-14-00293-f003:**
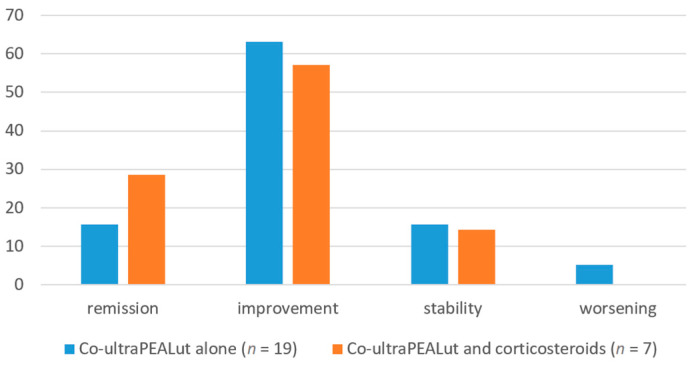
Subjective assessment of health change over time. Data displayed as %.

**Table 1 brainsci-14-00293-t001:** Patients’ characteristics.

	Co-ultraPEALut	Control
	(*n* = 26)	(*n* = 15)
Age (years)	54.69 ± 11.26	55.8 ± 14.84
Sex		
Female	19 (73.08%)	7 (46.67%)
Male	7 (26.92%)	8 (53.33%)
Years of education	14.10 ± 3.05	14.33 ± 3.46
Occupational status		
Employed	21 (80.77%)	12 (80%)
Unemployed	1 (3.85%)	0
Retired	3 (15.38%)	3 (20%)
COVID-19 severity		
Paucisymptomatic infection, at-home management	22 (84.62%)	14 (93.33%)
Hospitalization without ventilation	4 (15.38%)	1 (6.67%)
ICU admission with ventilation	0	0
NS-T0 interval (months)	7.33 ± 4.75	10.43 ± 5.12
T0-T1 interval (months)	9.92 ± 5.37	7.54 ± 2.36

Data displayed as mean ± standard deviation or n° (%). ICU: intensive care unit. NS: nasopharyngeal swab testing positive for SARS-CoV-2. T0: first neurological evaluation time. T1: re-evaluation time.

**Table 2 brainsci-14-00293-t002:** Outcome measures comparison between co-ultraPEALut-receiving patients and the control sample.

Co-ultraPEALut (*n* = 26)			
Test or Questionnaire	T0	T1	*p* Value
MoCA (raw score)	25.76 ± 2.3	27.2 ± 2	0.0261 *
MoCA (adjusted score)	24.34 ± 2.58	25.59 ± 2.39	0.0958
FSS	5.03 ± 1.69	4.52 ± 1.63	0.3188
PRMQ	51.94 ± 10.55	39.67 ± 13.02	<0.00001 *
SCL-90-R (T scores)			
DEP	1.52		
ANX	1.16		
**Control (*n* = 15)**			
**Test or questionnaire**	**T0**	**T1**	***p* value**
MoCA (raw score)	25.93 ± 2.6	-	-
MoCA (adjusted score)	24.74 ± 3.26	-	-
FSS	4.95 ± 1.57	4.06 ± 1.47	0.1352
PRMQ	45.77 ± 13.47	42.33 ± 16.86	0.2051
SCL-90-R (T scores)			
DEP	1.42		
ANX	1.17		

Data displayed as mean ± standard deviation. FSS: Fatigue Severity Scale. MoCA: Montreal Cognitive Assessment test. PRMQ: Prospective and Retrospective Memory Questionnaire. T0: first neurological evaluation time. T1: re-evaluation time. SCL-90-R: Symptom Checklist-90-Revised questionnaire. DEP: depression dimension of the SCL-90-R questionnaire. ANX: anxiety dimensions of the SCL-90-R questionnaire. *: statistical significance for *p* < 0.05.

**Table 3 brainsci-14-00293-t003:** Outcome measures comparison between the subsample treated with co-ultra-PEALut alone and the one treated with both co-ultra-PEALut and corticosteroids.

Co-ultraPEALut Alone (*n* = 19)			
Test	T0	T1	*p* Value
MoCA (raw score)	25.93 ± 2.52	27.11 ± 1.82	0.1417
MoCA (adjusted score)	24.54 ± 2.82	25.33 ± 2.03	0.8488
FSS	4.54 ± 1.54	4.29 ± 1.68	0.6803
PRMQ	53.38 ± 9.92	40.24 ± 13.3	0.0067 *
**Co-ultraPEALut and corticosteroids (*n* = 7)**			
**Test**	**T0**	**T1**	***p* value**
MoCA (raw score)	25.33 ± 1.21	27.43 ± 2.3	0.0709
MoCA (adjusted score)	24.47 ± 2.2	26.27 ± 3.11	0.2609
FSS	6.8 ± 0.23	5.24 ± 1.32	0.6251
PRMQ	47.25 ± 11.35	38.29 ± 12.88	0.2782

Data displayed as mean ± standard deviation. FSS: Fatigue Severity Scale. MoCA: Montreal Cognitive Assessment test. PRMQ: Prospective and Retrospective Memory Questionnaire. T0: first neurological evaluation time. T1: re-evaluation time. *: statistical significance for *p* < 0.05.

## Data Availability

The original contributions presented in the study are included in the article, further inquiries can be directed to the corresponding author.

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
