# Peer review of "Co-ultraPEALut in Subjective Cognitive Impairment Following SARS-CoV-2 Infection: An Exploratory Retrospective Study"

_brainsci, 2024, doi:10.3390/brainsci14030293_

Round 1

Reviewer 1 Report

Comments and Suggestions for Authors

Thank you for the opportunity to review the authors' manuscript. In order to improve the understanding of a wide range of readers, I would like to request the following corrections.

Please elaborate on the definition of subjective cognitive impairment for this study. Also, please explain why you selected the "Cognitive test" used in this study. In particular, it would help the reader understand better if you could explain why you did not select the Trail Making Test (TMT), which assesses attention disorders, or the Word fluency test (WFT), which assesses language fluency.

About the patients. Given the age of the target population, I believe it is necessary to investigate not only their educational history but also their occupational history.

About statistical analysis. Please provide the sample size and the results of the power calculations. Also state the statistical software and version used.

About Figure1. The pie chart does not describe the number of people, only the percentages are presented. I believe this is confusing to the reader.

This paper lacks content regarding "Study limitation". Without clarification of this point, I believe that the bias of the results obtained in this study cannot be denied.

Without clarification of these points, it is difficult to determine the consistency of the authors' views.

That is all. 

Please consider revising it.

Author Response

Dear Reviewer,

My co-authors and I wish to thank you for the precious comments and suggestions that we received on the submission entitled "Co-ultraPEALut in subjective cognitive impairment following SARS-CoV-2 infection: an exploratory retrospective study". We recognize that the suggestions and useful comments have greatly improved the paper’s scientific quality and impact. We have addressed your concerns below and have amended the manuscript according to your recommendations.

Suggested amendments that have been included in the manuscript are highlighted in yellow. Please, find our answers (in blue) to the comments below.

Please elaborate on the definition of subjective cognitive impairment for this study. Also, please explain why you selected the "Cognitive test" used in this study. In particular, it would help the reader understand better if you could explain why you did not select the Trail Making Test (TMT), which assesses attention disorders, or the Word fluency test (WFT), which assesses language fluency.

>> Thank you for the interesting comment.

We have added the definition of subjective cognitive impairment in the Materials and Methods section.

We chose not to include the Trail Making Test and the World Fluency Test in the battery of cognitive tests used in this study because MoCA already includes short versions of both, thus providing a measure of executive functioning in these domains; moreover, MoCA is one of the most widely used cognitive screening tests and it is of proved efficacy in the context of post-COVID cognitive impairments1.

1: B. Biagianti et al., ‘Cognitive Assessment in SARS-CoV-2 Patients: A Systematic Review’, Front. Aging Neurosci., vol. 14, Jul. 2022, doi: 10.3389/fnagi.2022.909661.

About the patients. Given the age of the target population, I believe it is necessary to investigate not only their educational history but also their occupational history.

>> Thanks for the suggestion. We have added the occupational status of the subjects in the main text and in Table 1.

About statistical analysis. Please provide the sample size and the results of the power calculations. Also state the statistical software and version used.

>> Our study is underpowered with respect to the power calculations performed. We have clarified this limitation in the Discussion section.

We have also added the statistical software used to the requested information, as we agree that this is an important detail to provide.

About Figure1. The pie chart does not describe the number of people, only the percentages are presented. I believe this is confusing to the reader.

>> This is a useful comment. We have changed Figure 1 to include the absolute numbers along with the percentages.

This paper lacks content regarding "Study limitation". Without clarification of this point, I believe that the bias of the results obtained in this study cannot be denied.

>> We agree that there are biases in the study, so we have better highlighted and expanded the limitations section.

Reviewer 2 Report

Comments and Suggestions for Authors

The submitted manuscript by Cenarcchi et al. discussed the effect of Co-ultraPEALut treatment on subjective cognitive impairment in COVID-19 patients. This is an exploratory retrospective study, not a randomized (RCT) trial. Overall the theme is interesting as a preliminary report, although a stronger and further study design (RCTs) is warranted. I did not see any serious flaw that requires major revision. I appreciate the acknowledgement of study limitations in the Discussion section. Something to note:

1) Pay attention to English editing (language), especially in the abstract. For example: better not to start a new sentence with a number (e.g. Line 17: 26 ......), but spell out "Twenty-six...".

2) Please highlight the significant contribution of this work as opposed to earlier studies (e.g. in Milan) that also used the same treatment method.

3) Do the authors plan to analyze anything on the MRI dataset? This may uncover neural mechanisms associated with the treatment.

Thanks.

Comments on the Quality of English Language

English editing is required. Multiple awkward sentences were found.

Author Response

Dear Reviewer,

My co-authors and I wish to thank you for the precious comments and suggestions that we received on the submission entitled "Co-ultraPEALut in subjective cognitive impairment following SARS-CoV-2 infection: an exploratory retrospective study". We recognize that the suggestions and useful comments have greatly improved the paper’s scientific quality and impact. We have addressed your concerns below and have amended the manuscript according to your recommendations.

Suggested amendments that have been included in the manuscript are highlighted in yellow. Please, find our answers (in blue) to the comments below.

1) Pay attention to English editing (language), especially in the abstract. For example: better not to start a new sentence with a number (e.g. Line 17: 26 ......), but spell out "Twenty-six...".

>> Thank you for the comment. We spelled out the number in the abstract and revised the entire manuscript for overall better English editing.

2) Please highlight the significant contribution of this work as opposed to earlier studies (e.g. in Milan) that also used the same treatment method.

>> This is a useful suggestion. We have added more references about the results that co-ultraPEALut has achieved in clinical studies on post-COVID patients.

3) Do the authors plan to analyze anything on the MRI dataset? This may uncover neural mechanisms associated with the treatment.

>> We agree that this would be a relevant further investigation. We do not plan to include MRI analysis in the present work, but we have included this suggestion in the paper as a possible future development.

English editing is required. Multiple awkward sentences were found.

Reviewer 3 Report

Comments and Suggestions for Authors

This is a very well written and interesting manuscript. However, I have some comments and suggestions which may improve the quality of this paper.

Introduction

  1. The introduction provides a good overview of post-COVID syndrome and the potential role of neuroinflammation. However, the focus on co-ultraPEALut as a treatment comes a bit abruptly. I would recommend providing more background on PEA and luteolin individually first before introducing their combination treatment.
  2. More information is needed on the purported mechanisms of action of PEA and luteolin in reducing neuroinflammation specifically. The introduction mentions their general anti-inflammatory and antioxidant properties but more details are needed on how these translate to effects on neuroinflammation.
  3. The third aim of the study comparing co-ultraPEALut alone to combination with corticosteroids seems tangential. I would recommend focusing the introduction more narrowly on the cognitive effects and comparison of co-ultraPEALut to untreated patients. The corticosteroid comparison seems like a secondary aim.
  4. The introduction would benefit from briefly mentioning what neuropsychological tests were used to assess cognitive function as an outcome measure. This connects the aims back to the methods.
  5. More background is needed on why this specific sample of post-COVID patients was chosen. Were they known to have cognitive impairment or was cognitive impairment prevalence being evaluated?

Methods

  1. More details are needed on the specific neuropsychological tests used, beyond just listing their names. Brief descriptions of what each test measures and the score interpretations would be helpful.
  2. The criteria for defining "subjective cognitive impairment" should be more clearly stated upfront. What specific symptoms or complaints qualified patients for inclusion? How severe did the impairment have to be?
  3. The exclusion criteria indicate patients with known cognitive or psychiatric disorders were excluded. However, depression and anxiety symptoms are common in post-COVID patients. Were questionnaires used to screen for these conditions?
  4. For the comparison group of patients not receiving co-ultraPEALut, were they matched to the treatment group on any demographic or clinical characteristics? If not, this should be stated as a limitation.
  5. In the corticosteroid subgroup analysis, what was the rationale for choosing this particular dosing regimen? How many patients received the combination treatment?
  6. Were there any adverse effects monitored with the treatments? If safety outcomes were not assessed this should also be stated.
  7. Statistical analysis section would benefit from stating what specific comparisons were made between groups and timepoints for each outcome measure.

Results

  1. The presentation of results is very clear overall. I would recommend organizing findings by outcome measure rather than participant group to better showcase the changes over time.
  2. The subjective health assessment results should be presented before comparisons to controls and corticosteroid groups. This keeps the within-group changes together.
  3. Statistical reporting can be streamlined. Phrasing such as "a statistically significant improvement was observed in MoCA raw scores from T0 to T1 (p=.026)" keeps the flow better than reiterating all the scores.
  4. In the control group comparisons, clearly state upfront that control patients did not have T1 assessments for the MoCA and subjective ratings. This contextualizes why statistical comparisons were not possible.
  5. The rationale behind comparing co-ultraPEALut alone vs with corticosteroids is unclear. This came out of nowhere in the introduction. Either foreshadow this comparison earlier or remove it if exploratory.
  6. Interpret the lack of differences between subgroups cautiously since sample sizes were highly uneven (n=19 vs n=7). The power to detect differences was likely limited.
  7. Consider presenting key results also in visual graphical format such as bar charts. this further highlights changes over time.

Discussion/Conclusion

In summary, the discussion and conclusion make important connections but need to avoid overstating the implications of this initial retrospective analysis. Maintaining measured interpretation and stating limitations/next steps will further enhance these sections.

Specifics:

  1. When comparing to control patients, caution is needed when stating the improvements were "not as impactful" based on the statistical test results. The differences between groups were not significant, meaning firm conclusions cannot be made about meaningful differences or lack thereof.
  2. Along those lines, the lack of statistically significant differences between the co-ultraPEALut alone and with corticosteroids groups should be interpreted more cautiously due to the small subsample sizes. Concluding they did not obtain "statistically different results" overstates the findings.
  3. Limitations around lack of randomization and blinding should be expanded upon. The results may reflect placebo effects rather than treatment effects without controlled conditions.
  4. The conclusion draws very definitive statements about co-ultraPEALut's beneficial role and suggested effect based on an uncontrolled, retrospective study. This overstates the implications. Rewording to indicate the findings "suggest" a potential effect and provide preliminarily evidence would be more accurate.
  5. Future research directions could be expanded beyond just using co-ultraPEALut with anti-inflammatories. Studies on mechanisms, randomized controlled trials, and specific neurological outcome evaluations could be interesting next steps.

Author Response

Dear Reviewer,

My co-authors and I wish to thank you for the precious comments and suggestions that we received on the submission entitled "Co-ultraPEALut in subjective cognitive impairment following SARS-CoV-2 infection: an exploratory retrospective study". We recognize that the suggestions and useful comments have greatly improved the paper’s scientific quality and impact. We have addressed your concerns below and have amended the manuscript according to your recommendations.

Suggested amendments that have been included in the manuscript are highlighted in yellow. Please, find our answers (in blue) to the comments below.

Introduction

1. The introduction provides a good overview of post-COVID syndrome and the potential role of neuroinflammation. However, the focus on co-ultraPEALut as a treatment comes a bit abruptly. I would recommend providing more background on PEA and luteolin individually first before introducing their combination treatment.

>> Thank you for the useful comment. We have added some background information on luteolin and PEA as single molecules to better introduce co-ultraPEALut, in addition to reorganizing and expanding the dedicated section in the Introduction paragraph.

2. More information is needed on the purported mechanisms of action of PEA and luteolin in reducing neuroinflammation specifically. The introduction mentions their general anti-inflammatory and antioxidant properties but more details are needed on how these translate to effects on neuroinflammation.

>> We have expanded the introduction in this regard and added more references about co-ultraPEALut's beneficial effects on neurological diseases/symptoms that have neuroinflammation as a putative etiopathogenetic mechanism.

3. The third aim of the study comparing co-ultraPEALut alone to combination with corticosteroids seems tangential. I would recommend focusing the introduction more narrowly on the cognitive effects and comparison of co-ultraPEALut to untreated patients. The corticosteroid comparison seems like a secondary aim.

>> Regarding your suggestion, we have removed the corticosteroid subgroup comparison from the aims and described it as a secondary exploratory analysis in the Methods section.

4. The introduction would benefit from briefly mentioning what neuropsychological tests were used to assess cognitive function as an outcome measure. This connects the aims back to the methods.

>> Regarding your suggestion, we have included in the Aims section the neuropsychological tests used, each accompanied by a brief description of the main domain tested, for greater consistency with the following Methods section.

5. More background is needed on why this specific sample of post-COVID patients was chosen. Were they known to have cognitive impairment or was cognitive impairment prevalence being evaluated?

>> We selected patients who were referred to our neurology clinic for post-COVID impairment who had a specific complaint of subjective cognitive impairment (usually attention, memory, lexical recall) after confirmed SARS-CoV-2 infection. Currently, there is no neuropsychological test score or accepted self- or observer/informant-based scale to classify an individual with subjective cognitive impairment, so we relied on a detailed clinical history.

We agree that expanding on the topic of subjective cognitive impairment would have been beneficial to the reader's understanding of our work, so we have added a more complete definition in the Materials and Methods section.

Methods

1. More details are needed on the specific neuropsychological tests used, beyond just listing their names. Brief descriptions of what each test measures and the score interpretations would be helpful.

>> Thank you for the comment. We have expanded the descriptions of the neuropsychological tests and questionnaires in the Methods section, as well as providing a general score interpretation for MoCA test.

2. The criteria for defining "subjective cognitive impairment" should be more clearly stated upfront. What specific symptoms or complaints qualified patients for inclusion? How severe did the impairment have to be?

>> Please see the answer to the previous comment in Introduction - point 5.

3. The exclusion criteria indicate patients with known cognitive or psychiatric disorders were excluded. However, depression and anxiety symptoms are common in post-COVID patients. Were questionnaires used to screen for these conditions?

>> This is an insightful comment. Patients with a previous diagnosis of cognitive or psychiatric disorders were excluded from our study; however, many included patients reported anxious and/or depressive symptoms after SARS-CoV-2 infection. We rephrased the section on exclusion criteria in our paper to better highlight this difference. In addition, we added the mean scores obtained in the depression and anxiety subscales of the Symptom Checklist-90-Revised (SCL-90-R) questionnaire administered at T0, which show that there are no significant differences at baseline between patients treated with co-ultraPEALut and controls.

4. For the comparison group of patients not receiving co-ultraPEALut, were they matched to the treatment group on any demographic or clinical characteristics? If not, this should be stated as a limitation.

>> We did not match patients receiving co-ultraPEALut with controls, although the two groups have similar demographic characteristics (see Table 1 in the main text). We have included this statement in the limitations section.

5. In the corticosteroid subgroup analysis, what was the rationale for choosing this particular dosing regimen? How many patients received the combination treatment?

>> Thank you for the useful observation.

We have corrected this data in the Methods section: the corticosteroid dosing regimen was 1 mg/kg/die instead of 50 mg/die as previously stated. The choice of this dosage was based on clinical judgment.

The number of patients who received the combination treatment is 7, as stated in the Results section.

6. Were there any adverse effects monitored with the treatments? If safety outcomes were not assessed this should also be stated.

>> Thank you for the interesting comment. We have added a statement in this regard in the Methods and Results sections.

7. Statistical analysis section would benefit from stating what specific comparisons were made between groups and timepoints for each outcome measure.

>> This is a useful insight. We have decided to report the statistical analysis performed once, at the beginning of the Materials and Methods section, so as not to overload the reading of the section. However, we have rearranged the Results section to clarify which comparisons were made for each outcome (see the response to the following comment in Introduction - point 1).

Results

1. The presentation of results is very clear overall. I would recommend organizing findings by outcome measure rather than participant group to better showcase the changes over time.

>> Regarding your recommendation, we have rewritten the Results section by reorganizing the findings and adding subheadings.

2. The subjective health assessment results should be presented before comparisons to controls and corticosteroid groups. This keeps the within-group changes together.

>> Thanks for the suggestion. We have rearranged the sentences accordingly.

3. Statistical reporting can be streamlined. Phrasing such as "a statistically significant improvement was observed in MoCA raw scores from T0 to T1 (p=.026)" keeps the flow better than reiterating all the scores.

>> Thanks for the suggestion. We have decided to keep the numerical results in the main text (in addition to listing them in the corresponding tables) for clarity; however, we have summarized the statistics in parentheses at the end of the sentences to improve readability.

4. In the control group comparisons, clearly state upfront that control patients did not have T1 assessments for the MoCA and subjective ratings. This contextualizes why statistical comparisons were not possible.

>> Thanks for the useful comment. The statement regarding lack of data at T1 for MoCA and at T0 for subjective assessment of change has been moved earlier in the text.

5. The rationale behind comparing co-ultraPEALut alone vs with corticosteroids is unclear. This came out of nowhere in the introduction. Either foreshadow this comparison earlier or remove it if exploratory.

>> We agree that the rationale was unclear in our previous version, so we have added a short paragraph and a reference in the Introduction section to better introduce our secondary analysis on corticosteroids.

6. Interpret the lack of differences between subgroups cautiously since sample sizes were highly uneven (n=19 vs n=7). The power to detect differences was likely limited.

>> Thanks for the comment. We have rephrased the paragraph in question.

7. Consider presenting key results also in visual graphical format such as bar charts. this further highlights changes over time.

>> Thanks for the suggestion. We have added a visual representation of the main results presented in Table 2 and in the main text in the form of a bar chart.

Discussion/Conclusion

In summary, the discussion and conclusion make important connections but need to avoid overstating the implications of this initial retrospective analysis. Maintaining measured interpretation and stating limitations/next steps will further enhance these sections.

Specifics:

1. When comparing to control patients, caution is needed when stating the improvements were "not as impactful" based on the statistical test results. The differences between groups were not significant, meaning firm conclusions cannot be made about meaningful differences or lack thereof.

>> We agree that the previous wording was inaccurate and have reworded the section in question so as not to imply significance in our comparisons. 

2. Along those lines, the lack of statistically significant differences between the co-ultraPEALut alone and with corticosteroids groups should be interpreted more cautiously due to the small subsample sizes. Concluding they did not obtain "statistically different results" overstates the findings.

>> As stated above, we agree with this comment. The section has been reworded accordingly, and a disclaimer about interpretation has been added.

3. Limitations around lack of randomization and blinding should be expanded upon. The results may reflect placebo effects rather than treatment effects without controlled conditions.

>> We have improved and expanded the limitation section based on your useful suggestions.

4. The conclusion draws very definitive statements about co-ultraPEALut's beneficial role and suggested effect based on an uncontrolled, retrospective study. This overstates the implications. Rewording to indicate the findings "suggest" a potential effect and provide preliminarily evidence would be more accurate.

>> We agree that our previous wording overstated the true impact of this study; we have revised the conclusion paragraph to better convey our findings.

5. Future research directions could be expanded beyond just using co-ultraPEALut with anti-inflammatories. Studies on mechanisms, randomized controlled trials, and specific neurological outcome evaluations could be interesting next steps.

>> Thank you for your suggestion. We have updated the section about future perspectives accordingly.

Round 2

Reviewer 1 Report

Comments and Suggestions for Authors

Thank you for your careful revisions.

I have the impression that it is easier to read than last time.

I believe the necessary corrections have been made.

Reviewer 3 Report

Comments and Suggestions for Authors

The authors responded to my comments very well. Thanks much.